# Cell Dome as an Evaluation Platform for Organized HepG2 Cells

**DOI:** 10.3390/cells12010069

**Published:** 2022-12-23

**Authors:** Ryotaro Kazama, Satoshi Fujita, Shinji Sakai

**Affiliations:** 1Department of Materials Engineering Science, Graduate School of Engineering Science, Osaka University, Toyonaka 560-8531, Japan; 2AIST-Osaka University Advanced Photonics and Biosensing Open Innovation Laboratory, National Institute of Advanced Industrial Science and Technology (AIST), Suita 565-0871, Japan

**Keywords:** HepG2 cell, Cell Dome, 3D culture, alginate, cell aggregates, spheroids, tissue engineering

## Abstract

Human-hepatoblastoma-derived cell line, HepG2, has been widely used in liver and liver cancer studies. HepG2 spheroids produced in a three-dimensional (3D) culture system provide a better biological model than cells cultured in a two-dimensional (2D) culture system. Since cells at the center of spheroids exhibit specific behaviors attributed to hypoxic conditions, a 3D cell culture system that allows the observation of such cells using conventional optical or fluorescence microscopes would be useful. In this study, HepG2 cells were cultured in “Cell Dome”, a micro-dome in which cells are enclosed in a cavity consisting of a hemispherical hydrogel shell. HepG2 cells formed hemispherical cell aggregates which filled the cavity of Cell Domes on 18 days of culture and the cells could continue to be cultured for 29 days. The cells at the center of hemispherical cell aggregates were observed using a fluorescence microscope. The cells grew in Cell Domes for 18 days exhibited higher Pi-class Glutathione S-Transferase enzymatic activity, hypoxia inducible factor-1α gene expression, and higher tolerance to mitomycin C than those cultured in 2D on tissue culture dishes (* *p* < 0.05). These results indicate that the center of the glass adhesive surface of hemispherical cell aggregates which is expected to have the similar environment as the center of the spheroids can be directly observed through glass plates. In conclusion, Cell Dome would be useful as an evaluation platform for organized HepG2 cells.

## 1. Introduction

Liver-derived cells are widely used in research, such as metabolite profiles and drug development [1], because it is important to investigate hepatotoxicity, hepatic metabolism, and absorption/excretion of candidate drugs in the liver [2]. The cell line most commonly used to study hepatotoxic mechanisms of drugs and toxins is HepG2 cells [3,4]. Since there are differences in the functional, transcriptomic, and proteomic levels between HepG2 cells cultured in 2D and primary human hepatocytes [5,6], HepG2 cells do not fully reflect all hepatic functions in vivo, although they are easier to handle than primary human hepatocytes and their responses are more reproducible [1,7].

In general, 2D substrates are widely used as a cell culture method and cell evaluation system for HepG2 cells, because of their simplicity, reproducibility, and low cost [8,9]. However, for drug development and mechanistic studies, evaluation using HepG2 cells cultured in 3D rather than those cultured in 2D is attracting attention, because cells cultured in 3D provide better biological models compared with those cultured in 2D [10,11,12]. HepG2 cells cultured in 3D exhibit enhanced expression of phase II drug metabolism enzymes and transporters [11,13,14]. Phase II enzymes are involved in detoxification and activation of many xenobiotics and transporters involved in their absorption and excretion. Their increased expression in HepG2 cells to the same level as that in primary hepatocytes is important for drug discovery and pharmacokinetic studies.

Based on the above studies, various 3D HepG2 cell culture systems have been reported, such as microwells [15,16], hanging drops [17], and microcapsules [18,19]. Previous 3D cell culture systems have been used to produce spherical spheroids. In HepG2 cells, the cells at the center of the spheroids become hypoxic and the expression of hypoxia-inducible factor-1α (HIF-1α), which is involved in various factors such as drug metabolism, is increased [20,21,22]. Until now, protocols such as fixation/sectioning or confocal laser scanning have been required to observe cells at the center of spheroids [23,24]. In this study, we aimed to evaluate the usefulness of a “Cell Dome” in the fabrication of organized hemispherical cell aggregates of HepG2 cells to easily observe the cut surface of the spheroid through glass plates. A platform that allows the easy observation and evaluation of HepG2 cells cultured in 3D would be useful.

We recently reported a culture/research system called the “Cell Dome”. The system was developed to culture non-adherent cells in 3D microenvironment [25]. Cell Dome is a hemispherical microdome fabricated on a glass plate in which cells are enclosed in a cavity consisting of a hemispherical hydrogel shell. After obtaining a hemispherical hydrogel containing cells by the thermal sol-gel transition of gelatin, a horseradish peroxidase (HRP)-mediated hydrogelation reaction was used to constitute a hemispherical alginate-based hydrogel shell. The hydrogel shells were immobilized on glass plates.

In this study, we evaluated the proliferation and behavior of HepG2 cells enclosed in Cell Dome, including cellular compatibility of the preparation process, changes in Pi-class Glutathione S-Transferase (GSTP1) activity, associated HIF-1α gene expression, and tolerance to antitumor drugs (mitomycin C), and compared them with those of cells cultured in 2D.

## 2. Materials and Methods

### 2.1. Materials

*N*,*N*-dimethylformamide (DMF), hydrogen peroxide aqueous solution (H_2_O_2_, 31 wt.%), catalase from bovine liver, horseradish peroxidase (HRP, 180 units/mg), mitomycin C, and *N*-hydroxysuccinimide (NHS) were obtained from Fujifilm Wako Pure Chemical Industries (Osaka, Japan). Aminosilane (APS)-coated glass plates were obtained from Matsunami Glass Ind., Ltd. (Osaka, Japan). The plates were 0.8 mm thick, 18 mm × 18 mm in size, and had a 6 × 6 array of water-repellent patterns printed in a ring shape with an outer diameter of 1.4 mm and an inner diameter of 1 mm (intervals: 2.5 mm). If necessary, the glass plates were cut to a suitable size and used in each experiment. Tyramine hydrochloride and 3-(4-hydroxyphenyl) propionic acid were obtained from Chem-Impex International (Wood Dale, IL, USA), and Tokyo Chemical Industry (Tokyo, Japan), respectively. Sodium alginate (Kimica I-1G, Mannuronic acid/Guluronic acid ratio ≈ 0.7) and water-soluble carbodiimide hydrochloride (WSCD·HCl) were purchased from Kimica (Tokyo, Japan) [26], and the Peptide Institute (Osaka, Japan), respectively. Fluorescein isothiocyanate-dextran (FITC-dextran), type A gelatin from porcine skin (ca. 300 g bloom), type B gelatin from bovine skin (ca. 250 g bloom), trypsin from porcine pancreas, and alginate lyase from *Flavobacterium* sp., were obtained from Sigma-Aldrich (St. Louis, MO, USA). Calcein-AM was purchased from Nacalai Tesque Inc. (Kyoto, Japan). Ethylenediamine-*N*,*N*,*N*’,*N*’-tetraacetic acid, disodium salt, dihydrate (EDTA·2NA), and propidium iodide (PI) were obtained from Dojindo (Kumamoto, Japan). The primers used for real time-PCR were purchased from Eurofins Genomics (Tokyo, Japan). 3-(4-Hydroxyphenyl)propionic acid *N*-hydroxysuccinimide ester (HPP-NHS) was synthesized using WSCD⋅HCl, NHS, and 3-(4-hydroxyphenyl) propionic acid as previously described [27].

Alginate-Ph was prepared by conjugating tyramine hydrochloride to sodium alginate via NHS/WSCD·HCl chemistry, and gelatin-Ph was prepared by conjugating 3-(4-hydroxyphenyl) propionic acid to type B gelatin via NHS/WSCD⋅HCl chemistry, as described previously [28,29]. The phenolic hydroxyl (Ph)-content of alginate-Ph and gelatin-Ph were 1.1 × 10^−4^ mol-Ph/g and 2.4 × 10^−4^ mol-Ph/g, respectively.

HepG2 cells were purchased from the Riken Cell Bank (Ibaraki, Japan) and grown in Dulbecco’s modified Eagle’s medium (DMEM; Nissui, Tokyo, Japan) supplemented with 10% (*v*/*v*) fetal bovine serum. The cells cultured in 2D were grown in cell culture dishes. The cells were cultured at 37 °C in an atmosphere humidified with 5% CO_2_ and 95% air.

### 2.2. Cell Enclosed in Cell Dome

Cell Domes enclosing HepG2 cells were prepared based on a previously reported method [25]: To immobilize phenolic hydroxyl on the APS-coated glass plates, the glass plates were immersed in DMF containing 2.5% (*w*/*v*) HPP-NHS overnight. Next, PBS (phosphate-buffered saline, pH 7.4, 1 µL) containing 50 U/mL HRP, 1.2 × 10^6^ cells/mL HepG2 cells, and 3.0% (*w*/*v*) type A gelatin was spotted at 5 °C in the water-repellent patterns printed in a ring shape on the glass plates and allowed to stand for 15 min. Then, PBS (10 µL) containing 1 mM H_2_O_2_, 1.0% (*w*/*v*) gelatin-Ph, and 0.5% (*w*/*v*) alginate-Ph were dropped onto the gelatin gel hemisphere at 15 °C. After hydrogel shell formation through HRP-mediated hydrogelation for 5 min, the remaining polymer-Ph solution was rinsed twice with PBS. The glass plates with Cell Domes were immersed in medium containing catalase at 0.5 mg/mL and replaced with a catalase-free medium after 24 h of incubation. The medium was replaced every 2–3 days. As gelatin gels transition from gel to sol with increasing temperature, the enclosed gelatin gel dissolves upon incubation at 37 °C, forming a hemispherical cavity in Cell Dome (Figure 1). HepG2 hemispherical cells aggregates cultured for 18 days were mainly used for analysis, because the cells fully fill the hemispherical cavity in Cell Dome. The viability of the enclosed HepG2 cells was measured using the fluorescence live/dead assay stained with Calcein AM and PI. Fluorescence microscopy (BZ-9000; Keyence, Tokyo, Japan) and was used to observe HepG2 cells in Cell Domes. Optical coherence tomography (Cell3iMager Estier, Screen, Kyoto, Japan) was used to visualize the cells grown three-dimensionally in Cell Dome.

### 2.3. Hydrogel Permeability

The diffusion coefficients of FITC-dextrans (M.W. 4000 and 70,000) in water (*D*_w_) and the hydrogel (*D*_gel_) were measured by fluorescence recovery after photobleaching (FRAP) at 37 °C based on a previous study [30]. A confocal laser scanning microscope (C2; Nikon, Tokyo, Japan) was used for the measurements. To prepare the hydrogel, PBS (180 µL) containing 1.0% (*w*/*v*) gelatin-Ph, 0.5% (*w*/*v*) alginate-Ph, and 50 U/mL HRP and PBS (20 µL) containing 1 mM H_2_O_2_ were mixed. The resulting hydrogel was immersed overnight at 37 °C in PBS containing FITC-dextran at 0.5 mg/mL. Fluorescent dyes in water or the hydrogel were bleached for a short time using a laser, and the fluorescence intensity of the FITC-dextran diffusing into the bleached area was measured. Based on the measured fluorescence intensity, *D*_w_ and *D*_gel_ were calculated. The permeability of the hydrogel shell was evaluated as *D*_gel_*/D*_w_.

### 2.4. GSTP1 Enzymatic Activity

GSTP1 enzymatic activity in HepG2 cells was analyzed by staining with GSTP1 Green (Funakoshi, Tokyo, Japan). Cell Domes were immersed in Hank’s balanced salt solution with 20 mM HEPES (pH 7.4, HBSS) containing 2.5 µM GSTP1 Green, and 10 µM MK571 (Funakoshi, Tokyo, Japan) at 37 °C for 30 min. The cells were washed twice with HBSS, and HepG2 cells stained with GSTP1 Green were observed and analyzed using fluorescence microscopy and flow cytometry (Accuri C6; BD Biosciences, Tokyo, Japan), respectively. The collection of single HepG2 cells for flow cytometry analysis was performed as follows, Cell Domes were soaked in PBS containing alginate lyase at 1 mg/mL for 5 min to degrade the alginate-based hydrogel shell. The hemispherical cell aggregates were then treated with trypsin-EDTA, and single cells were collected.

### 2.5. HIF-1α Gene Expression

To stain the HepG2 cells with Hypoxia Probe Solution: LOX-1 (MBL, Nagoya, Japan), cellular hypoxia in the cells was analyzed. Cell Domes were immersed in media containing a 2.0 µM LOX-1 at 37 °C for 24 h. Then, they were washed twice with PBS, and the HepG2 cells stained with the LOX-1 were observed using a fluorescence microscope. The relative amount of HIF-1α mRNA was determined by real-time PCR using specific primers (Table 1). Total RNA of the cells was extracted using the CellAmp Direct TB Green RT-qPCR Kit (Takara Bio, Shiga, Japan) according to the manufacturer’s instructions, and single-stranded cDNA was prepared by a PCR reaction that proceeded as follows: 37 °C for 30 min and 85 °C for 5 s. From the obtained cDNA template, the target sequences were amplified and quantified by real-time PCR using the specific primers. The PCR reaction was conducted as follows: 95 °C for 30 s and 40 cycles of 95 °C for 5 s and 60 °C for 10 s. GAPDH was used as the internal reference gene. Threshold cycle (Ct) values were measured and calculated using a real-time PCR system (CFX Connect™, Bio-Rad Laboratories, Hercules, CA, USA). The relative mRNA content was calculated as x = 2^−ΔCt^ where ΔCt = Ct_HIF-1α_ − Ct_GAPDH_. Experiments were performed in triplicate and data were presented as the mean of relative quantities of mRNA. For real-time PCR, HepG2 cells in Cell Domes after 18 days of culture were collected using the same methods described in Section 2.4.

### 2.6. Drug Treatment and Proliferation Inhibition

Cell Dome was fabricated on a glass plate cut to have one water-repellent ring pattern and was placed in one well of a 48-well plate. The resultant glass plate with one Cell Dome was immersed in a solution of PBS (15 μL) containing 1–1000 nM mitomycin C mixed with medium (150 μL) and incubated for 72 h. Then, the reagent (15 µL) for evaluating the mitochondrial activity of cells (Cell Count Reagent SF, Nacalai Tesque, Kyoto, Japan) was added to each well. After incubating for 4 h, the absorbance of the supernatant at 450 nm was measured using a UV-VIS spectrophotometer (UV-2600, Shimadzu, Kyoto, Japan). The “relative activity” was defined using the following formula as the value proportional to the number of viable cells. *A*_treated_ and *A*_control_ in the following formula show the average increase in the absorbance value of the supernatant per Cell Dome after addition/no addition of mitomycin C, respectively.
Relative activity (%) = (1 − *A*_treated_/*A*_control_) × 100(1)

### 2.7. Statistical Analysis

Data were presented as the mean ± standard deviation (SD). Statistical differences in two datasets and three datasets were analyzed using a Student’s *t*-test and one-way analysis of variance (ANOVA) followed by post hoc using Tukey’s analysis, respectively.

## 3. Results

### 3.1. Hydrogel Permeability

The *D*_gel_ of FITC-dextrans with M.W. 4000 and 70,000 in the hydrogels prepared at the same composition as the shell of Cell Domes were 2.9 × 10^−10^ m^2^/s, and 8.7 × 10^−11^ m^2^/s, respectively, and their *D*_gel_/*D*_w_ values were 0.91 ± 0.18, and 0.58 ± 0.14, respectively. *D*_w_ and *D*_gel_ were defined as the diffusion coefficients of the solute in water and the hydrogel, respectively.

### 3.2. Cell Growth in Cell Dome

The viability of HepG2 cells before enclosing, immediately after closing, and one day after enclosing were 98.6 ± 0.8%, 96.6 ± 1.1%, and 97.9 ± 1.0%, respectively (Figure 2a). Immediately after enclosing, the HepG2 cells in Cell Domes existed individually (Figure 2b). The enclosed cells began to form aggregates after one day of culture (Figure 2c). The area occupied by the cells increased with the increasing culture period (Figure 2d–f). Cell Domes maintained their shape over 29 days of culture (Figure 2g). This result indicates that HepG2 cells proliferated in Cell Domes. The growth of the enclosed cells was also observed as an increase in mitochondrial activity, corresponding to an increased number of living cells per Cell Dome (Figure 2h). As shown in Figure 2i, the enclosed HepG2 cells grew three-dimensionally and formed hemispherical cell aggregates that filled the cavity of Cell Domes on 18 days of culture (Figure 2i).

### 3.3. GSTP1 Enzymatic Activity

Figure 3a shows HepG2 cells in Cell Domes stained with GSTP1 Green after 18 days of culture. The green fluorescence intensity observed on the enclosed cells corresponding to the higher GSTP1 activity was higher than that observed on 2D-cultured cells on a tissue culture dish (Figure 3b). Flow cytometric measurement also confirmed this tendency (Figure 3c), and HepG2 cells collected from Cell Domes after 18 days of culture had higher GSTP1 activity than 2D-cultured cells on tissue culture dishes (Figure 3c). The mean fluorescence intensities of HepG2 cells not stained with GSTP1 Green, 2D-cultured cells on tissue culture dishes, and HepG2 cells collected from Cell Domes after 18 days of culture were 4.1 × 10^3^ ± 0.6 × 10^3^ [a.u.], 5.7 × 10^4^ ± 1.1 × 10^4^ [a.u.], and 2.3 × 10^5^ ± 0.5 × 10^5^ [a.u.], respectively (*n* = 3, * *p* < 0.05).

### 3.4. Expression of HIF-1α

Figure 4a shows HepG2 cells cultured in Cell Domes for 18 days and stained with LOX-1. Red fluorescence attributed to hypoxia was observed in the cells at the center of hemispherical cell aggregates formed by Cell Dome after 18 days of culture. After 18 days of culture, the hydrogel shells consisting of Cell Domes were degraded by soaking in PBS containing alginate lyase, and hemispherical HepG2 spheroids cultured in Cell Dome were collected within 5 min. HepG2 cells collected from Cell Domes after 18 days of culture exhibited a 4.8 ± 1.0-fold higher expression of the HIF-1α gene (Figure 4b, * *p* < 0.05).

### 3.5. Drug Treatment of Enclosed Cells

Figure 5 shows the relative activity of 2D-cultured HepG2 cells and the cells in Cell Dome after 1 or 18 days of culture and after 72 h of exposure to mitomycin C. The relative activities are defined as values proportional to the number of viable cells. HepG2 cells in Cell Dome after 1 d of culture and 2D-cultured cells exhibited similar relative activity (Figure 5). In contrast, hemispherical cell aggregates formed after 18 days of culture in Cell Dome exhibited higher relative activity than the 2D-cultured cells (* *p* < 0.05).

## 4. Discussion

Organized tumor cells obtained by culturing tumor cells in 3D better reflect the pathophysiology of tumor tissues than those cultured in 2D [33,34]. This is due to the abundance of cell–cell interactions compared with 2D-cultured cells and the presence of biochemical concentration gradients in organized tissues similar to tumor tissue in vivo. The observation and analysis of the cells in the hypoxic region resulting from the oxygen concentration gradient are important in tumor studies because these cells play a significant role in tumorigenesis [22,35,36].

Organized HepG2 cells including spheroids are useful in studies related to liver cancer, including drug discovery [5,13,37]. Therefore, various 3D cell culture methods have been reported to obtain HepG2 cell spheroids [38,39]. Cultivations in spinner flasks [40], microwells [15,16], ultralow attachment dishes [41], and the hanging drop method [17] have been widely used for this purpose. However, it is difficult to control the size and shape of spheroids using these methods when culturing with appropriate medium changes. The size and shape of organized HepG2 cells can affect the outcome and efficiency of drug discovery [42,43]. Thus, a 3D culture system that enables control of the size and shape of organized cells is required [43]. Cultivation in uniform microcapsules composed of semipermeable hydrogel membranes is a promising method for obtaining spheroids of controlled sizes [18,19]. Tumor cells grow and fill the cavities of the microcapsules, and the size of the spheroids can be controlled by controlling the cavity size. Not only are the spheroids formed in the microcapsules, but the spheroids formed through various methods are limited by the need for confocal imaging for the observation and analysis of the hypoxic region in the center of spheroids [23,24,44]. In this study, we expected that cell domes would allow us to obtain hemispherical HepG2 cell aggregates with easily observable hypoxic regions induced by an oxygen concentration gradient. The enclosed HepG2 cells grew and filled the cavity and formed a hemispherical aggregate templated by the cavity (Figure 2i). In addition, the existence of a hypoxic region at the center of the glass adhesive surface of the aggregate was confirmed by using conventional fluorescence microscopy, as expected (Figure 4a).

The hydrogel shell of Cell Domes was made of a polymer mixture of alginate-Ph and gelatin-Ph, which is known for its high cell compatibility, and was formed using an HRP-mediated hydrogelation reaction [45]. HRP-mediated hydrogelation has been used in the preparation of cell-laden hydrogels, such as microcapsules [18,46] and constructs fabricated by 3D bioprinting [47,48]. Measurement of the diffusion coefficient of FITC-dextran (*D*_gel_) in the hydrogel revealed that the hydrogel mixture of alginate-Ph and gelatin-Ph prepared through HRP-mediated hydrogelation exhibited high permeability to low-molecular-weight compounds. This indicates that oxygen and nutrients necessary for the cells enclosed in Cell Domes to live and grow permeate well and that metabolites discharged by the cells also diffuse from the inside of Cell Domes to the external medium. In fact, HepG2 cells proliferated in Cell Dome during 29 days of culture (Figure 2b–g). In addition, the permeability of the hydrogel allowed the evaluation of HepG2 cells enclosed in Cell Dome using fluorescence dye supplied from the surrounding solution to evaluate cell viability (Figure 2a, and Appendix A), cellular GSTP1 activity (Figure 3), and cellular hypoxia (Figure 4a). A limitation of permeable hydrogel shells is that the cells enclosed in the adjacent Cell Dome are stained with the same reagents supplied by the medium. This could be solved if the staining reagents are supplied individually from inside the Cell Dome.

An attractive feature of the alginate-based hydrogel shell is that it enables the collection of enclosed cells simply by soaking the solution containing alginate lyase for evaluating the cells using flow cytometry (Figure 3c) and real-time PCR (Figure 4b). In contrast, when using alginate-based hydrogels, we must be careful in selecting the reagents that are supplied to cells inside the hydrogel shell because alginate is a negatively charged polysaccharide [49]. We do not have data, but to avoid adsorption of positively charged reagents, it would be effective to use other polymers such as chitosan-Ph, a phenolic hydroxyl modified with chitosan, which dissolves at a neutral pH, to prepare the hydrogels by HRP-mediated reactions [50].

Regarding the stability, HepG2 cell-laden Cell Domes maintained their shape after the cells filled the hemispherical cavities after 18 days of culture, as indicated by the microscope image shown in Figure 2g and the constant value of the mitochondrial activity per Cell Dome (Figure 2h). Toxicity testing and drug development require long-term, low-level exposure to test compounds; however, traditional screening systems such as primary human hepatocyte cells cultured in 2D can only survive for a few days [5,51]. The ability to culture cells for long periods would allow various studies to be evaluated over a long period, which may be useful for drug development. The period during which the hemispherical cavity is filled by the enclosed cells would be controlled by adjusting the concentration of cells initially enclosed and the size of Cell Domes. In addition, the size of hemispherical cell aggregates would be controlled by controlling the size of the cavity.

HepG2 cells formed hemispherical cell aggregates in Cell Domes and showed higher drug resistance than those cultured in 2D on tissue culture dishes (Figure 5). Similar results have been reported for HepG2 cells cultured in a previous 3D culture system using 3D cell culture hydrogel [52] and microwells [16]. Zhou et al. also reported that HepG2 spheroids cultured in 3D showed higher tolerance to mitomycin C than 2D-cultured cells [53]. These results could be explained by the higher GSTP1 enzymatic activity and higher HIF-1α gene expression in the hemispherical aggregates of HepG2 cells. Glutathione S-transferases (GST) are multifunctional enzymes that play important roles in cellular detoxification. As GSTs can react with the thiol moiety of glutathione, almost all types of compounds are the substrate for GSTs [2]. They are excreted extracellularly by multidrug resistance-associated protein (MRP) transporters. There are various types of GSTs, among which Pi class GST (GSTP1) is highly expressed in various cancers and contributes significantly to the acquisition of drug resistance in cancer cells [54]. Therefore, the up-regulation of GST and GSTP1 in liver cancer promotes the extracellular metabolism of antitumor drugs and reduces their efficacy [55]. Ohkura et al. reported that human hepatocyte spheroids cultured for seven days had higher GSTP1 mRNA expression than those cultured for two days [56]. Based on these findings, it is important to investigate GSTP1 enzymatic activity in HepG2 cell aggregates to determine whether it can be applied in the study of candidate compounds for new drug development. Additionally, in vivo tumors have an oxygen concentration gradient and hypoxia occurs in tumors [21,33]. Hypoxic cells overexpress hypoxia-inducible factor-1 (HIF-1), which increases the expression of genes that contribute to drug resistance [35]. Fu et al. showed that the expression of HIF-1α in HepG2 spheroids formed using silk fibroin after five days of culture (3D) was significantly higher than that in cells cultured for one day (2D) [57]. Fu et al. also showed that the viability of HepG2 spheroids treated with antitumor drugs was significantly higher than that of 2D-cultured cells [57]. In fact, HepG2 cells at the center of hemispherical cell aggregates formed by Cell Dome after 18 days of culture, were observed with red fluorescence attributed to hypoxia and the expression of HIF-1a in the hemispherical cell aggregates was higher than that in 2D-cultured cells (Figure 4). These results indicated the center of the hemispherical cell aggregates is hypoxic and the existence of a gradient of oxygen concentration in Cell Dome. Additionally, the GSTP1 activity of HepG2 cells collected from Cell Dome was higher than that of 2D-cultured cells on tissue culture dishes (Figure 3, * *p* < 0.05). Flow cytometry analysis of cells collected from Cell Dome and stained with GSTP1 Green showed a small peak immediately before the fluorescence peak was observed (Figure 3c). This peak would indicate a non-hypoxic cell population in the vicinity of the hydrogel shell with low GSTP1 enzyme activity, due to the oxygen concentration gradient in hemispherical cell aggregates. Based on these findings, the HepG2 hemispherical cell aggregates were resistant to mitomycin C in this study. Although a more detailed mechanistic analysis of the tolerance of the enclosed cells to mitomycin C was outside the scope of this study, gene expression of MRP (ABCC1), multidrug resistance (MDR; ABCB1), ATP-binding cassette transporter G2 (ABCG2), and other genes induced by increased HIF-1α gene expression may contribute to the acquisition of drug resistance in enclosed cells [58,59].

## 5. Conclusions

In this study, HepG2 cells were cultured in a “Cell Dome” to form a hemispherical cell aggregate that adhered to glass plates. The cells at the center of the glass adhesive surface of the hemispherical cell aggregate, which showed specific characteristics attributed to hypoxia, could be observed without using a confocal laser imaging microscope. These results demonstrate the feasibility of Cell Dome as an evaluation platform of organized HepG2 cells.

## Figures and Tables

**Figure 1 cells-12-00069-f001:**
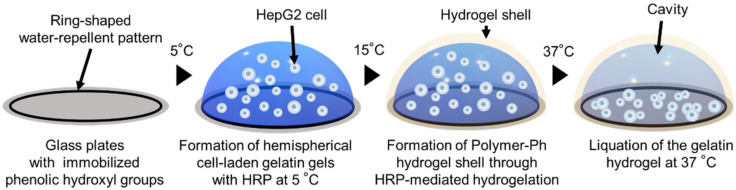
Schematic drawing of Cell Dome preparations on glass plates. Cells are enclosed in a cavity consisting of hemispherical hydrogel shells.

**Figure 2 cells-12-00069-f002:**
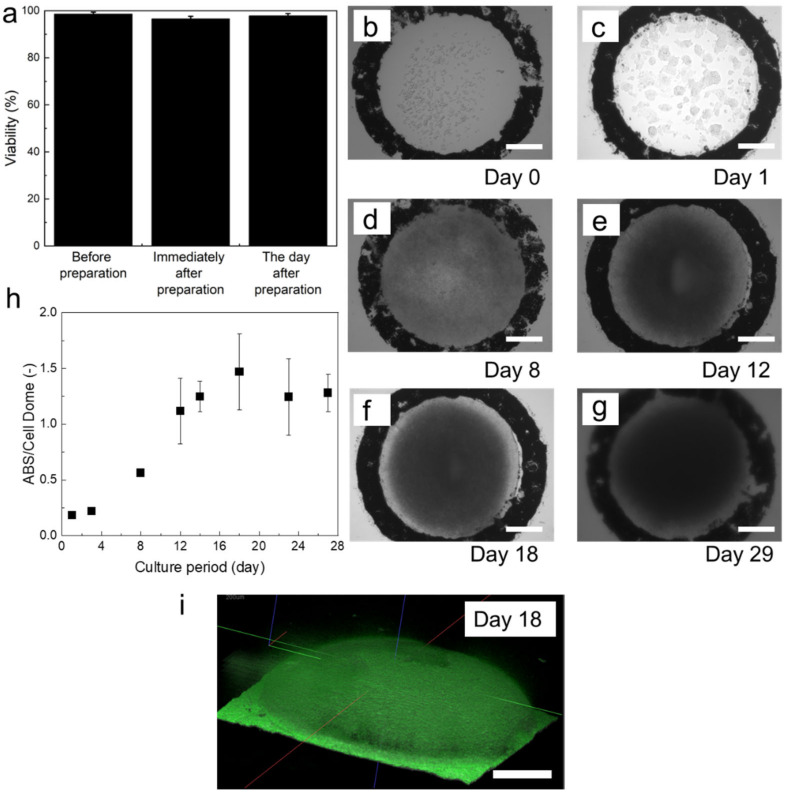
(**a**) HepG2 viabilities before enclosing and, immediately and a day after enclosing. (**b**–**g**) HepG2 cell growth in Cell Dome, (**h**) the mitochondrial activity, corresponding to a number of living cells per Cell Dome, (**i**) optical coherence tomography image of HepG2 cells after 18 days of culture. Bars in a and h: SD, *n* = 5, Bars in (**b**–**g**) and (**i**): 250 µm.

**Figure 3 cells-12-00069-f003:**
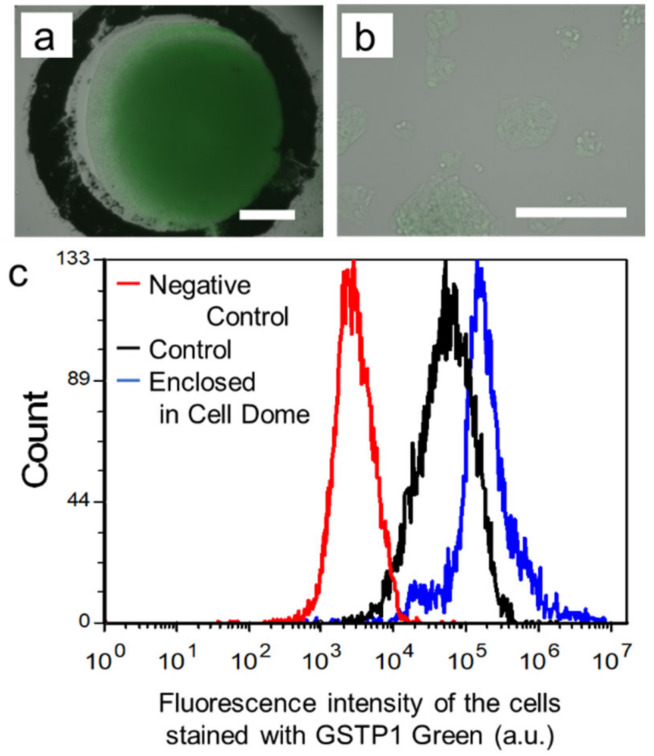
(**a**,**b**) Fluorescence observation of HepG2 cells in Cell Domes after 18 days of culture (**a**) and 2D-cultured cells on a tissue culture dish (**b**) both stained with GSTP1 Green. Bars: 250 µm. (**c**) Flow cytometry analysis data of the cells not stained with GSTP1 Green (Red line: Negative Control) and stained with GSTP1 Green enclosed in Cell Domes after 18 days of culture (Blue line) and 2D-cultured cells on a tissue culture dish stained with GSTP1 Green (Black line: Control). * *p* < 0.05 vs. 2D-cultured cells with and without staining GSTP1 Green.

**Figure 4 cells-12-00069-f004:**
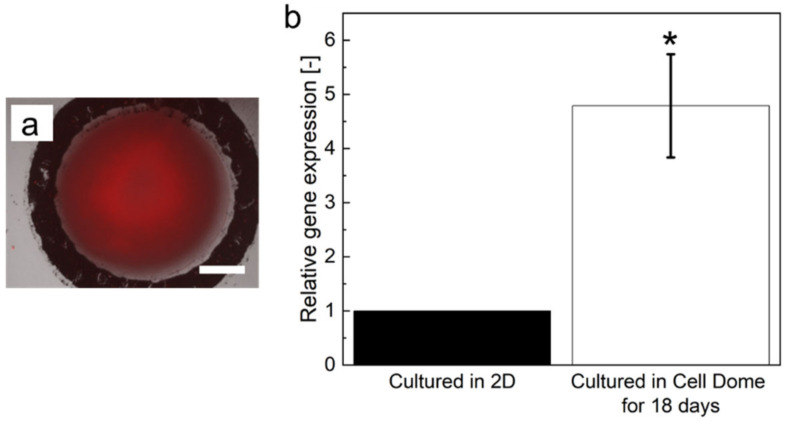
(**a**) Fluorescence observation of HepG2 cells cultured in Cell Domes at 18 days and stained with LOX-1. Bar: 250 µm. (**b**) HIF-1α gene expressions in 2D-cultured HepG2 cells on a tissue culture dish and those cultured in Cell Domes for 18 days. Each value was normalized by the means of data in the 2D-cultured cells. Bars: SD, (*n* = 3). * *p* < 0.05 vs. 2D-cultured cells on a tissue culture dish.

**Figure 5 cells-12-00069-f005:**
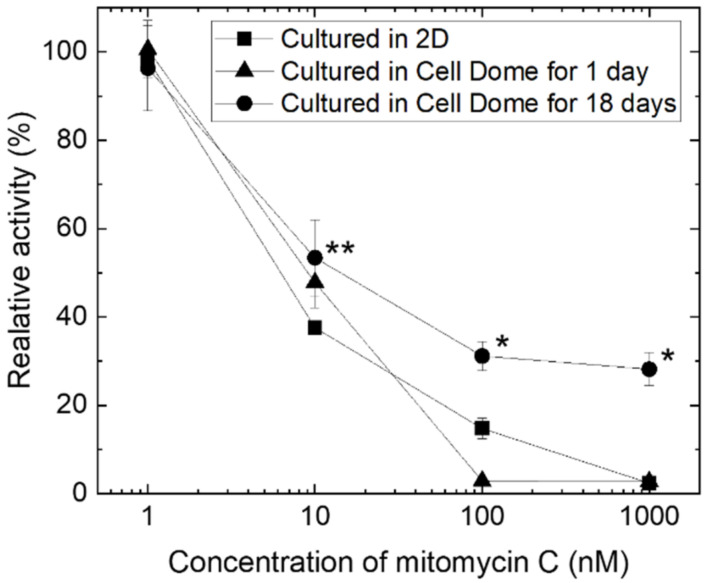
Relative activity of 2D-cultured cells on a tissue culture dish (■), the cells in Cell Dome after 1 day of culture (▲), and hemispherical cell aggregates formed in Cell Dome after 18 days of culture (●). Bars: SD, *n* = 3~5, ** p* < 0.05 vs. 2D-cultured cells and those in Cell Dome after 1 day of culture, ** *p* < 0.05 vs. cultured in 2D-cultured cells.

**Table 1 cells-12-00069-t001:** Primer used for detecting HIF-1α gene expression.

Gene	Primer Forward	Primer Reverse	Reference
GAPDH	5′-GGA GTC CCT GCC ACA CTC AG-3′	5′-GGC CCC TCC CCT CTT CA-3′	[31]
HIF-1α	5′-TGC ATC TCC ATC TCC TAC CC-3′	5′-CCT TTT CCT GCT CTG TTT GG-3′	[32]

## Data Availability

The data that support the findings of this study are available upon reasonable request from the authors.

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
