# Peer review of "Cell Dome as an Evaluation Platform for Organized HepG2 Cells"

_cells, 2022, doi:10.3390/cells12010069_

Round 1
Reviewer 1 Report
In the manuscript entitled “Cell Dome as an evaluation platform for organized HepG2 cells”, authors evaluated the proliferation and behavior of HepG2 cells enclosed in Cell Dome and concluded that is useful as a platform for research and evaluation of HepG2 cells. Authors demonstrated that Cell Dome is a promising tool as an evaluation platform for organized HepG2 cells.
My overall evaluation of the manuscript is positive. The manuscript is well-structured and develops on the subject stated in the title. Before it can be accepted for publication there are a number of revision, formal and scientific aspects that should be addressed.
1) Abstract should be rewritten based on “IMRaD” format.
2) In Abstract section, Authors should be mentioned the relevant statistical significant and tests performed.
3) In order comparison, Authors should be mentioned the mean intensity (MFI) in figure of flowcytometry test.
4) Please make the discussion more explain in terms of comparison with other similar studies. The limitations of the study should be elaborated in the discussion section.
Author Response
We appreciate the reviewer for their insightful feedback on ways to strengthen our paper. We have incorporated changes that reflect the detailed suggestions and comments the reviewers have provided.
Response to Reviewer 1
In the manuscript entitled “Cell Dome as an evaluation platform for organized HepG2 cells”, authors evaluated the proliferation and behavior of HepG2 cells enclosed in Cell Dome and concluded that is useful as a platform for research and evaluation of HepG2 cells. Authors demonstrated that Cell Dome is a promising tool as an evaluation platform for organized HepG2 cells.
My overall evaluation of the manuscript is positive. The manuscript is well-structured and develops on the subject stated in the title. Before it can be accepted for publication there are a number of revision, formal and scientific aspects that should be addressed.
1) Abstract should be rewritten based on “IMRaD” format.
Reply) Thank you for your suggestion. We have added short paragraphs for discussion and conclusion on the abstract according to your suggestion (Page 1, Line 10-29).
2) In Abstract section, Authors should be mentioned the relevant statistical significant and tests performed.
Reply) According to your suggestion, we have added the relevant statistical significance in the Abstract section (Page 1, Line 25).
3) In order comparison, Authors should be mentioned the mean intensity (MFI) in figure of flowcytometry test.
Reply) Thank you for the comment. We have added the results of mean intensity (MFI) in flow cytometry analysis (Page 6, Line 243-246).
4) Please make the discussion more explain in terms of comparison with other similar studies. The limitations of the study should be elaborated in the discussion section.
Reply) We have added a discussion in terms of comparison with other similar studies (Page 10, Line 361, 362, 372-374, 380-384). We also have added a limitation of the study in the discussion section (Page 9, Line 318-321).
Finally, we thank the reviewer for giving us the opportunity to strengthen our manuscript with valuable comments and queries.
Reviewer 2 Report
The manuscript by Kazama et al. reports the application of the "Cell Dome" technique on HepG2 cells. The "Cell Dome" method was developed by the authors and the submitted study describes the characterization of cultivated HepG2 cells in a hemispherical hydrogel for up to 29 days. HepG2 cells enclosed in the Cell Domes were easy to observe and showed a better functionality compared to classical 2D tissue culture.
There are several issues that need to be addressed:
Introduction:
3D vs 2D vs primary hepatocytes: Please, could authors consider to rewrite the paragraph between line 38 - 44 describing the significance of Phase II enzymes in the comparison of different cells and culture methods. Because of the complexity of the information it is hard to follow.
The aim of the study needs to be specified (alternative...) in the introduction section.
Please check on the citation #25 since the paper is not published yet, could the authors provide a DOI?
Comparison to 2 D culture should be mentioned - line 63-66
Materials and Methods
Please check the Journals requirements regarding the numeration.
Please consider to split the Materials in different paragraphs or assign the materials to the specific method.
GSTP1:
Please could authors provide the information why MK571, the inhibitor of MDR, was used.
HIF
Please could authors comment on the gene expression study: why was analyzes performed on day 18?
Mitomycin C treatment
Please could authors comment on the Mitomycin C treatment, why was Mitomycin C selected? Is there an clinical context?
Results:
Permeability
Please could authors consider to add the significance of the obtained result, though it is discussed later.
cell growth:
Please could authors add a negative control to Fig 3c, flow cytometry and add a comment about the cell population right before the fluorescence peak in dome cells fraction. Please quantify the activity of GSTP1 and please add the statistic.
gene expression HIF 1alpha:
Please could the authors interpret the significant increase on the gene expression of HIF 1 alpha in cell dome cultivated HepG2 cells compared to cells cultivated 2D.
Drug treatment:
Please could the authors interpret the significant higher viability? of cell dome cultivated HepG2 cells compared to cells cultivated 2D after treatment.
Minor:
Method for 2D cultivation is missing, please add
line 190 please check space
line 196 please check space
Author Response
We appreciate the reviewers for their insightful feedback on ways to strengthen our paper. We have incorporated changes that reflect the detailed suggestions and comments the reviewers have provided.
Response to Reviewer 2
The manuscript by Kazama et al. reports the application of the "Cell Dome" technique on HepG2 cells. The "Cell Dome" method was developed by the authors and the submitted study describes the characterization of cultivated HepG2 cells in a hemispherical hydrogel for up to 29 days. HepG2 cells enclosed in the Cell Domes were easy to observe and showed a better functionality compared to classical 2D tissue culture.
There are several issues that need to be addressed:
Introduction
1) 3D vs 2D vs primary hepatocytes: Please, could authors consider to rewrite the paragraph between line 38 - 44 describing the significance of Phase II enzymes in the comparison of different cells and culture methods. Because of the complexity of the information it is hard to follow.
Reply) Thank you for your suggestion. As you pointed out, the paragraph between lines 38-44 was complex, so we have rewritten this paragraph (Page 1, line 37-42, Page 2, Line 48, 49, 53, 54).
2) The aim of the study needs to be specified (alternative...) in the introduction section.
Reply) Thank you for your valuable comments. We have rewritten the description to clarify the purpose of the study in the introduction section based on your suggestion (Page 2, Line 62).
3) Please check on the citation #25 since the paper is not published yet, could the authors provide a DOI?
Reply) Thank you for your comment. We have added information on paper of the citation #25 to the manuscript and below (Page 12, Line 474).
#25 Development of non-adherent cell-enclosing domes with enzymatically cross-linked hydrogel shell 2023 Biofabrication, 15 015002 https://doi.org/10.1088/1758-5090/ac95ce.
4) Comparison to 2D culture should be mentioned - line 63-66
Reply) Thank you for your suggestion. We have rewritten the description of lines 63-66 based on your suggestion (Page 2, Line 78, 79).
Materials and Methods
1) Please check the Journals requirements regarding the numeration.
Reply) Thank you for pointing out our mistake. We have rewritten the numeration in the Materials and Methods section.
2) Please consider to split the Materials in different paragraphs or assign the materials to the specific method.
Reply) Considering your advice, we have split the Materials into different paragraphs (Pages 2, 3).
3) GSTP1: Please could authors provide the information why MK571, the inhibitor of MDR, was used.
Reply) Thank you for your valuable comments. We used the MRP transporter inhibitor MK571 based on the manufacturer's recommendation, because glutathione-conjugate which is conjugated by glutathione and GSTP1 Green (Funakoshi, Tokyo, Japan) may be excreted out of the cell by the MRP transporter, in accordance with the original foreign body exclusion mechanism of GST (Glutathione S-Transferase).
4) HIF: Please could authors comment on the gene expression study: why was analyzes performed on day 18?
Reply) We greatly thank for this comment. HIF analysis was performed on day 18 because the enclosed HepG2 cells formed hemispherical cell aggregates on day 18 of the culture.
5) Mitomycin C treatment: Please could authors comment on the Mitomycin C treatment, why was Mitomycin C selected? Is there an clinical context?
Reply) Thank you for your comments. We used mitomycin C in this study because mitomycin C has been used to evaluate the tolerance to an anti-cancer drug of HepG2 spheroids [44, REF1, 2].
[REF1] B. t, Braak et al., Towards an advanced testing strategy for genotoxicity using image-based 2D and 3D HepG2 DNA damage response fluorescent protein reporters, Mutagenesis, 37, (2022), 130–142.
[REF2] O. Sirenko et al., High-Content Assays for Characterizing the Viability and Morphology of 3D Cancer Spheroid Cultures, MARY ANN LIEBERT, INC. 13(7), (2015), 402-414. https://doi.org/10.1089/adt.2015.655.
Results:
1) Permeability: Please could authors consider to add the significance of the obtained result, though it is discussed later.
Reply) Thank you for your suggestion. We already provided several explanations related to permeability (Page 9, Line 303-326). And we have added that the permeability of the hydrogel shells allowed the staining of the enclosed cells with Hypoxia Probe; LOX-1 (Medical & Biological Laboratories, Nagoya, Japan) (Page 9, Line 317, 318).
2) cell growth: Please could authors add a negative control to Fig 3c, flow cytometry and add a comment about the cell population right before the fluorescence peak in dome cells fraction. Please quantify the activity of GSTP1 and please add the statistic.
Reply) Thank you for your suggestion. We have added a negative control to Fig 3c, statical analysis, and a comment about the cell population right before the fluorescence peak in dome cells fraction (Page 7 Figure 3c Line 253, 254, and Page10 Line 391-395).
3) gene expression HIF 1alpha: Please could the authors interpret the significant increase on the gene expression of HIF 1 alpha in cell dome cultivated HepG2 cells compared to cells cultivated 2D.
Reply) Thank you for your comments. HIF-1α is activated when the intracellular environment is hypoxic [REF3]. We interpreted that HepG2 cells in the center of hemispherical cell aggregates cultured in Cell Dome were hypoxic with a concomitant significant increase in HIF-1α gene expression (Page 10, Line 377-389).
[REF3] K. Balamurugan, HIF-1 at the crossroads of hypoxia, inflammation, and cancer, Int. J. Cancer, 138(5) (2017) 1058-1066. https://doi.org/10.1002%2Fijc.29519
4) Drug treatment: Please could the authors interpret the significant higher viability? of cell dome cultivated HepG2 cells compared to cells cultivated 2D after treatment.
Reply) Thank you for your valuable comments. The significantly higher viability of hemispherical HepG2 cell aggregates cultured in Cell Dome compared to the cells cultured in 2D after treatment would be explained by the higher GSTP1 enzyme activity and HIF-1α gene expression in hemispherical cell aggregates. The up-regulation of GST and GSTP1 in liver cancer promotes the extracellular metabolism of antitumor drugs and reduces their efficacy [46] and hypoxic cells overexpress hypoxia-inducible factor-1 (HIF-1), which increases the expression of genes that contribute to drug resistance [48] (Page 10, Line 377-389).
Minor:
1) Method for 2D cultivation is missing, please add.
Reply) Thank you for pointing out our mistake. We have appended a method for 2D culture (Page 3, Line 113, 114).
2) line 190 please check space
Reply) Thank you for pointing out our mistake. Thanks for the heads up, I have corrected the space in line 190 (Page 5, Line 213).
Finally, we thank the reviewers for giving us the opportunity to strengthen our manuscript with valuable comments and queries.
Reviewer 3 Report
Major comments
The title of this research is not clearly stated the use of the Cell Dome for what purpose. Although the results show a few characteristics and anticancer drug sensitivity of the Cell Dome, it lacks data supporting the potential use of this platform for drug discovery in comparison with the other spheroid, e.g., handing drop.
The term of a spheroid is based on a spherical shape. If the cell aggregate exhibits a hemispherical shape, it should be aware of the definition of the spheroid. Thus, 3D cell aggregate would be more appropriate to call the enclosed HepG2 in the Cell Dome.
The authors emphasize direct observation of cells in the hypoxic center of the spheroid is difficult without using a confocal laser scanning microscope. However, there is no data show advantage of the Cell dome for visualizing the hypoxic center of the Cell Dome.
Here is the list of experiments that need to be performed in order to show the advantages of the new 3D platform.
1) Compare the hemispherical Cell Dome with another spheroid in terms of 1.1 Expressions of phase II drug metabolism enzymes and transporters
1.2 Pi-class Glutathione S-Transferase (GSTP1) activity
1.3 HIF-1 protein expressions
1.4 Tolerance to mitomycin C
2) Fig. 2(a) HepG2 viabilities before enclosing and, immediately and a day after enclosing. Authors need to show representative fluorescence images of live and dead cells.
3) For Fig. 3C, it needs to show three-independent experiments and statistical test.
Author Response
We appreciate the reviewers for their insightful feedback on ways to strengthen our paper. We have incorporated changes that reflect the detailed suggestions and comments the reviewers have provided.
Response to Reviewer 3
The title of this research is not clearly stated the use of the Cell Dome for what purpose.
Reply) Thank you for your valuable comments. We have rewritten the description to clarify the purpose of the study in the introduction section based on your suggestion (Page 2, Line 62).
Although the results show a few characteristics and anticancer drug sensitivity of the Cell Dome, it lacks data supporting the potential use of this platform for drug discovery in comparison with the other spheroid, e.g., handing drop.
Reply) In this paper, we found that hemispherical cell aggregates formed by Cell Dome showed
(a) Higher Glutathione S-Transferase (GSTP1) enzymatic activity whose up-regulation promotes the extracellular metabolism of antitumor drugs and reduces their efficacy.
(b) Higher Hypoxia Inducible Factor-1α (HIF-1α) gene expressions and the center of the hemispherical cell aggregates are hypoxic.
(c) Higher tolerance to mitomycin C and it is suggested that drug/toxin excretion mechanisms such as transporters are enhanced
Thus, Cell Dome we reported has the same advantages as conventional 3D culture methods (e.g., spheroid and hanging drop methods). In addition, Cell Dome has the specific advantage of allowing direct observation of cells at the center of cell aggregates.
Cell Dome, which allows direct observation of the specific behavior caused by hypoxia in the center of cell aggregates, could provide new evidence for drug discovery and development.
The term of a spheroid is based on a spherical shape. If the cell aggregate exhibits a hemispherical shape, it should be aware of the definition of the spheroid. Thus, 3D cell aggregate would be more appropriate to call the enclosed HepG2 in the Cell Dome.
Reply) As you pointed out that the term spheroid is based on a spherical shape, so the name, which was defined as a hemispherical spheroid formed by Cell Dome, has been redefined as a hemispherical cell aggregate.
The authors emphasize direct observation of cells in the hypoxic center of the spheroid is difficult without using a confocal laser scanning microscope. However, there is no data show advantage of the Cell dome for visualizing the hypoxic center of the Cell Dome.
Reply) As you pointed out that there is no data show advantage of the Cell dome for visualizing the hypoxic center of the Cell Dome, so we have added Figure 4a images of enclosed cells stained with Hypoxia Probe: LOX-1 (MBL: Medical & Biological Laboratories, Nagoya, Japan) and observed by using a fluorescence microscope (Page 4 Line 169-173, Page 7 Line 257-259, and Page 8 Figure 4a).
Here is the list of experiments that need to be performed in order to show the advantages of the new 3D platform.
1) Compare the hemispherical Cell Dome with another spheroid in terms of
1.1 Expressions of phase II drug metabolism enzymes and transporters
Reply) Thank you for your suggestion. Pi-class Glutathione S-Transferase (GSTP1) is one of the phase II drug metabolism enzymes, so we analyzed GSTP1 activity as an example of phase II drug metabolism enzymes, though it is discussed later in 1.2. The activity of ATP Binding Cassette (ABC) transporters, including multidrug resistance-related protein (MRP) and multidrug resistance (MDR), was increased by increased HIF-1α gene expression [50, 51], and hemispherical spheroids formed by Cell Dome in this study increased HIF-1α expression compared to 2D cultured cells on tissue culture dishes (Page 8, Figure 4b).
1.2 Pi-class Glutathione S-Transferase (GSTP1) activity
Reply) Thank you for your comment. Pi-class Glutathione S-Transferase (GSTP1) activity in spheroids of human hepatocytes was reported in the previous report, so we have added a discussion (Page 10, Line 372-374).
1.3 HIF-1 protein expressions
Reply) Thank you for your suggestion. Expressions of HIF-1α in HepG2 spheroids were reported in the previous report, so we have added a discussion (Page 10, Line 380-384).
1.4 Tolerance to mitomycin C
Reply) Thank you for your comment. Tolerance to mitomycin C in HepG2 spheroids was reported in the previous report, so we have added a discussion (Page 10, Line 361, 362).
2) Fig. 2(a) HepG2 viabilities before enclosing and, immediately and a day after enclosing. Authors need to show representative fluorescence images of live and dead cells.
Reply) Thanks for your valuable comment. We have added representative fluorescent images of live and dead cells in HepG2 viability before inclusion, immediately after inclusion, and 1 day after inclusion (Fig. S1).
3) For Fig. 3C, it needs to show three-independent experiments and statistical test.
Reply) We have added three-independent experiments and statistical tests in figure 3c (Page 7, Figure 3c).
Finally, we thank the reviewers for giving us the opportunity to strengthen our manuscript with valuable comments and queries.
Round 2
Reviewer 2 Report
The revised manuscript by Kazama was significantly improved and only minor considerations should be addressed:
Please could authors consider to add a reason why the cell domes were cultivated until day 29 while experiments ended on day 18? Maybe because of the stagnation in mitochondrial activity from day 18 on as seen from the diagramm Fig 2h? Is there an statistical relavance? Please comment on this in the manuscript.
line 390 please check That
Author Response
Thank you for taking the time to review our manuscript and for your valuable feedback. We have modified our manuscript based on the comments.
1) Please could authors consider to add a reason why the cell domes were cultivated until day 29 while experiments ended on day 18? Maybe because of the stagnation in mitochondrial activity from day 18 on as seen from the diagramm Fig 2h? Is there an statistical relavance? Please comment on this in the manuscript.
Reply) Thank you for the comment. Considering your advice, we have rewritten statements about the reason why the cell domes were cultivated until day 29 (Page 1, Line 17-18, Page 3, Line 122-124, and Page 5, Line 220.).
2) line 390 please check That
Reply) Thank you for pointing out our mistake. Thanks for the heads up, I have corrected “That” in line 390 (Page 5, Line 371).
Finally, we thank reviewer 2 for giving us the opportunity to strengthen our manuscript with valuable comments and queries. We have worked hard to incorporate the feedback and hope that our revised submission is now acceptable for publication in “Cells”.
Reviewer 3 Report
Thanks for attempting to respond to the comments. The authors provided additional data to support the use of the cell dome as an alternative platform for drug discovery. To propose the cell dome as such, it needs more data in comparison to other 3D models. Considering the data sufficiency and scientific soundness, I still keep my previous decision of rejection. I hope this decision will not discourage the authors from further improving the manuscript.
Author Response
Thank you for taking the time to review our manuscript and for your valuable feedback.
Response to Reviewer 3
Thanks for attempting to respond to the comments. The authors provided additional data to support the use of the cell dome as an alternative platform for drug discovery. To propose the cell dome as such, it needs more data in comparison to other 3D models. Considering the data sufficiency and scientific soundness, I still keep my previous decision of rejection. I hope this decision will not discourage the authors from further improving the manuscript.
Reply) Thank you for the comment. We will keep your comments in mind for future reference.
Finally, we thank reviewer 3 for giving us the opportunity to strengthen our manuscript with valuable comments and queries. We have worked hard to incorporate the feedback and hope that our revised submission is now acceptable for publication in “Cells”.